# Methionine Dependence of Cancer

**DOI:** 10.3390/biom10040568

**Published:** 2020-04-08

**Authors:** Peter Kaiser

**Affiliations:** Department of Biological Chemistry, School of Medicine, University of California, Irvine, CA 92697, USA; pkaiser@uci.edu; Tel.: +1-949-824-9367

**Keywords:** methionine, *S*-adenosylmethionine, cell cycle, cancer, SAM-checkpoint

## Abstract

Tumorigenesis is accompanied by the reprogramming of cellular metabolism. The shift from oxidative phosphorylation to predominantly glycolytic pathways to support rapid growth is well known and is often referred to as the Warburg effect. However, other metabolic changes and acquired needs that distinguish cancer cells from normal cells have also been discovered. The dependence of cancer cells on exogenous methionine is one of them and is known as methionine dependence or the Hoffman effect. This phenomenon describes the inability of cancer cells to proliferate when methionine is replaced with its metabolic precursor, homocysteine, while proliferation of non-tumor cells is unaffected by these conditions. Surprisingly, cancer cells can readily synthesize methionine from homocysteine, so their dependency on exogenous methionine reflects a general need for altered metabolic flux through pathways linked to methionine. In this review, an overview of the field will be provided and recent discoveries will be discussed.

## 1. Introduction

In 1959, Sugimura and colleagues reported a study where tumor-bearing rats were fed diets with the restriction of individual essential amino acids. Tumor growth was significantly affected by a methionine-restricted diet [1]. The dependence of cancer cell proliferation on methionine was further highlighted in 1973 by experiments that showed leukemia cells cannot proliferate in growth media where methionine is substituted with its metabolic precursor, homocysteine [2]. A flurry of experiments during the 70s and 80s expanded the methionine/homocysteine substitution experiments to many cell lines derived from various tumor sites. The results unequivocally established that the vast majority of cancer cells cannot proliferate when methionine is replaced with homocysteine, but non-cancer cells are indifferent to such replacement [3,4,5,6,7]. Table 1 lists cell lines with a known status of methionine dependence. This metabolic phenomenon that differentiates cancer cells from non-tumor cells is often referred to as the methionine dependence of cancer, the methionine stress sensitivity of cancer, or the Hoffman effect (Figure 1). It soon became clear that the addiction of cancer cells to exogenously provided methionine is not due to their failure to synthesize methionine from homocysteine [3,8], but is likely caused by increased demand for metabolites derived from methionine [5,7,8,9]. While mechanisms behind the Hoffman effect are yet to be fully understood, progress has been made towards this goal.

## 2. Methionine Metabolism

Methionine is an essential amino acid in mammals. In addition to its role as a component of proteins, methionine links to a number of important metabolic pathways that play key roles in epigenetics (*S*-adenosylmethionine), nuclear functions (polyamines), detoxification (glutathione), and cellular membranes (phospholipids) (Figure 2). Furthermore, the methionine cycle is intimately linked with folate metabolism and thus can indirectly modulate nucleotide biosynthesis.

Methionine is obtained through the diet. It is converted to the principal cellular methyl donor, *S*-adenosylmethionine (SAM, also referred to as AdoMet), through the transfer of adenosine from ATP to the methionine sulfur. This reaction is catalyzed by methionine adenosyl transferases (MAT). Mat1A is the main transferase in the liver, whereas extrahepatic tissues rely on the Mat2A/Mat2B complex for SAM synthesis. Mat2A is the catalytic subunit, but binding of the regulatory subunit Mat2B modifies kinetic properties by decreasing the K_m_ for methionine and sensitizing the enzyme to product inhibition [14,15,16]. SAM is used as a cofactor in most methylation reactions and provides the activated methyl group for conjugation to proteins, DNA, and lipids. *S*-adenosylhomocysteine (SAH) remains after the methyl group transfer and is hydrolyzed into homocysteine and adenosine by SAH hydrolase. Homocysteine is then remethylated using 5-methyltetrahydrofolate as the methyl group donor to regenerate methionine and complete the methionine cycle (Figure 2A,B). Notably, vitamin B_12_ is required to transfer methyl groups from 5-methyltetrahydrofolate to remethylate homocysteine (Figure 2B). In the liver and kidney, homocysteine can also be remethylated using betaine, which is derived from choline. However, most other tissues do not express the necessary enzyme, betaine–homocysteine methyltransferase [17,18], and rely on the folate and B_12_-mediated remethylation reaction.

Maintaining the necessary methylation potential in cells via generation of SAM is a major role of methionine metabolism. However, one needs to consider that SAH is a potent inhibitor of methyltransferases [19], and the ability of cells to methylate substrates is thus not only determined by SAM abundance, but also by SAH levels. The cellular methylation potential is thus best expressed as the SAM/SAH ratio.

## 3. Polyamine Synthesis and Methionine Metabolism

Besides its function as a methylation cofactor, SAM is also the sole donor of aminopropyl groups in polyamine synthesis [20]. Polyamines are essential for cell growth and enzymes involved in their synthesis are often overexpressed in cancer [21]. Polyamines are required at relatively high concentrations in cells (millimolar range), and thus need a substantial amount of SAM to maintain homeostasis during cell proliferation [22]. Polyamine synthesis is initiated by decarboxylation of SAM to form dcSAM, which serves as the aminopropyl donor for spermine and spermidine synthase. dcSAM is converted into 5′-deoxy-5′-methylthioadenosine (MTA) after donating the aminopropyl group. MTA is then processed in the methionine salvage pathway through multiple steps to recycle adenine and methionine (Figure 2A). Maintaining flux through this cycle is critical for two reasons. First, circumstantial evidence suggests that dcSAM inhibits DNA methyl transferases [23], and perhaps other methyltransferases. Second, MTA is a competitive inhibitor of protein arginine N-methyltransferase 5 (PRMT5) [24]. PRMT5 is overexpressed in many cancers and correlates with patient survival [25]. The significance of MTA as a PRMT5 inhibitor becomes evident when we consider the rate limiting enzyme of the methionine salvage pathway, methylthioadenosine phosphorylase (MTAP). MTAP is frequently deleted in tumors due to its proximity to the tumor suppressor gene CDKN2A [26,27,28,29]. Consequently, tumor cells with MTAP deletions have elevated levels of MTA, which acts as a SAM competitive inhibitor for the PRMT5 methyltransferase, but not other methyltransferases [24]. A potential therapeutic angle of this phenomenon was exposed in a synthetic lethal approach that identified PRMT5 as a vulnerability of cancers with MTAP deletion, because the elevated MTA levels in these cancers already limit necessary PRMT5 activity [24]. Normal cells or cancers with intact MTAP, and thus high flux through the methionine salvage pathway, are less sensitive to PRMT5 inhibition. These studies highlight an unexpected connection of methionine metabolism, polyamine synthesis, and cancer, which may contribute to the Hoffman effect.

Note that even though flux through polyamine synthesis needs to be high, especially in dividing cells, neither the methionine cycle nor polyamine synthesis consumes methionine. However, methionine is removed by protein synthesis and indirectly through the transsulfuration pathway leading to cysteine synthesis and feeding antioxidant production (Figure 2A).

## 4. Methionine Metabolism and Cancer

Methionine metabolism has been connected to cancer on several levels. This review focuses on the Hoffman effect, which describes the dependence of cancer cells on exogenous methionine. Most cancer cells cannot proliferate in medium where methionine is replaced by homocysteine, even though they readily synthesize methionine from homocysteine (Figure 1). In fact, when intracellular methionine levels were measured in breast cancer cells after they had been shifted to homocysteine medium, methionine levels remained largely constant [8]. Non-cancer cells are indifferent to replacement of methionine with homocysteine. Differential metabolic dependencies of cancer and normal cells are often difficult to interpret, because different growth rates of cancer and normal cells can indirectly lead to distinct metabolic needs. However, the existence of several rapidly proliferating methionine-independent cancer cell lines argues against a major influence of proliferation rate (Table 1) [5,6]. Why some cancer cells remain methionine independent is not well understood. In addition, more direct approaches have definitely excluded indirect effects from growth rate-related metabolic dependence [7,8,30,31,32]. Most notably, when methionine-dependent cancer cells are continuously cultured in homocysteine medium, very rare cell clones can be selected that reverted to methionine independence without a change in proliferation rate [7,8,30,31,32]. Remarkably, most of these cancer cell-derived methionine-independent clones have lost properties associated with the tumorigenic state [8,30]. The mechanism of reversion is not known, but chromosomal alterations have been correlated with reversion to methionine independence in some systems [31]. However, reversion of MDA-MB468 triple negative breast cancer cells seems to be an epigenetic event, because the reverted state is semi-stable and needs to be stabilized occasionally by growth in homocysteine medium [32]. As indicated above, generation of methionine stress resistant cancer cells is usually coupled with loss of tumorigenic properties such as the ability of anchorage independent growth and proliferation in 1% serum. Conversely, oncogenic mutations in phosphoinositide 3-kinase (PI3K) and H-ras expression have been shown to promote methionine dependence [10,33]. Whether other oncogenes induce a similar metabolic dependence has not been systematically investigated, but these experiments indicate a tight link between methionine dependence and tumorigenicity. These findings are also encouraging considering clinical implications. Treatment strategies exploiting the methionine dependence of cancer should not be easily overcome by tumors developing resistance to methionine restriction, because these cancer cell line experiments indicate that tumorigenic properties are lost when cancer cells escape methionine dependence.

Even though cancer cells readily synthesize methionine from homocysteine and show similar intracellular steady-state methionine levels as methionine-independent cells, they cannot proliferate in these conditions. There are indications of a qualitative difference between exogenous and homocysteine-derived methionine. Such a different utilization of exogenous and synthesized methionine has been noted in double-label experiments, where cells preferentially incorporated exogenous methionine [34]. The reason for this difference is unknown. One issue to consider is that steady-state metabolite measurements do not inform us about flux through metabolic pathways and can therefore be misleading. For example, methionine levels may be kept constant at the expense of flux into downstream pathways. Indeed, tracing experiments in MDA-MB468 breast cancer cells with labelled homocysteine revealed a diversion of flux from the methionine cycle to the transsulfuration pathway (Figure 2A) [8]. These same experiments indicated the reduced synthesis of SAM, resulting in a reduced overall methylation potential reflected in a lower SAM/SAH ratio. These tracing experiments suggested that low SAM or SAM/SAH ratios are key to understanding the Hoffman effect, and that increased flux through the transsulfuration pathway may induce these changes in methylation potential. Notably, none of these metabolic effects were observed in revertant, methionine-independent MDA-MB468-R8 clones. The differential metabolic fates of homocysteine in methionine-dependent (transsulfuration) and -independent cells (mainly methionine cycle) observed in breast cancer cell systems may be a general feature of the Hoffman effect. Indeed, similar redirection of homocysteine into the transsulfuration branch is induced by oncogenic PI3K mutations, which induce a methionine-dependent phenotype when expressed in methionine-independent cells (Figure 2A) [10]. It is interesting to note that the transsulfuration pathway is largely restricted to pancreas, liver, and kidney, but is active in cancer cells [8,35,36,37], which could contribute to the differential response of cancer cells to growth in homocysteine medium. One can speculate that higher demand for antioxidants like glutathione in cancer may require increased flux through the transsulfuration pathway. However, at least in MDA-MB468 breast cancer cells, supplementation with antioxidants cannot compensate for methionine dependence [38].

Effects of methionine substitution with homocysteine on cellular methylation potential have been observed previously [3], but the significance of SAM as an important mediator of the Hoffman effect became clear when SAM supplementation of homocysteine medium eliminated the growth defect of MDA-MB468 cells [7]. Accordingly, reducing SAM synthesis directly by knockdown of MAT2A/B without affecting methionine levels resulted in a similar cell cycle arrest as observed by methionine replacement [9]. The realization that SAM is one of the key metabolites related to the methionine dependence of cancer may also connect other metabolic pathways important to cancer with the Hoffman effect. For example, the serine–glycine biosynthesis pathway is critical to maintaining cellular methylation potential (Figure 2B). This pathway maintains flux through the folate cycle. Thereby, serine to glycine conversion sustains homocysteine remethylation to methionine, but also stimulates nucleotide biosynthesis, including ATP, which, together with methionine, provides the building blocks for SAM (Figure 3) [39]. Several cancer cell lines and tumors depend on exogenous serine. This is surprising because serine can usually be synthesized from glucose in sufficient amounts via the glycolysis intermediate 3-phosphoglycerate [40,41]. However, the increased demand of cancer cells on glycolysis for energy production may divert the flux of 3-phosphoglycerate from serine synthesis to glycolysis. Serine requirement for cancer cell proliferation may thus link the Warburg and Hoffman effects by connecting glycolytic flux with SAM synthesis.

## 5. A Metabolic Cell Cycle Checkpoint Related to Methionine Metabolism

The cellular methylation potential (SAM/SAH ratio), and SAM abundance in particular, have been identified as a major contributor to the Hoffman effect [7]. Reduced cellular methylation potential is observed when cancer cells are grown in homocysteine medium or when SAM synthesis is reduced by knockdown of MAT2A/B [7,8,9]. These conditions induce a specific cell cycle arrest followed by apoptosis if the metabolic deficiency is not resolved. Such behavior is reminiscent of cell cycle checkpoint events, which are pathways that trigger cell cycle arrest to ensure cellular integrity under stress conditions [42]. Prolonged cell cycle check point arrests typically lead to apoptosis to protect organismal health. A prototypic cell cycle checkpoint is the response to DNA damage. Cells induce cell cycle arrest when DNA is damaged to allow time for repair and to maintain genetic stability. If the damage is not repaired, apoptosis is induced to remove these cells with potentially compromised genomes [43]. We have proposed a similar concept for maintenance of epigenetic stability [7,9,44,45]. This metabolic checkpoint is proposed to measure the cellular methylation potential (SAM/SAH ratio) and stop the cell cycle if the SAM/SAH ratio is too low to assure faithful duplication of epigenetic marks in the form of DNA and histone methylation during cell proliferation. We propose that the Hoffman effect is a manifestation of this “SAM-checkpoint”. In support of this concept, an evolutionarily conserved cell cycle arrest has been observed in response to SAM or methionine depletion from yeast to mammals (Figure 4). For cancer cells, but not normal cells, a shift to homocysteine medium is sufficient to trigger the SAM-checkpoint [7,9,44,45,46,47].

## 6. The SAM Checkpoint in Yeast

Yeast cells show a robust cell cycle arrest when SAM levels or methionine levels are reduced [44,46,47,48] (Figure 4). A special role of methionine metabolism in the cell cycle distinct from that of other amino acids was already noted in 1976 by Nobel laureate Lee Hartwell using the yeast model [49]. Many years later, a dedicated signaling pathway that connects methionine metabolism to the cell cycle in yeast was defined [44,46,47,48]. At the center of this intersection between methionine metabolism and cell cycle is the ubiquitin ligase SCF^Met30^. This ubiquitin ligase ubiquitylates two major substrates, the transcription factor Met4 and the cell cycle inhibitor Met32 [46,47,50]. The mechanism by which Met32 inhibits the cell cycle is not understood, but Met32 stabilization blocks *S*-phase initiation even when cyclin-dependent kinase activity is high [44,51]. Furthermore, the cell cycle arrest caused by the loss of SCF^Met30^ function is completely suppressed by deletion of *MET32* [46,50,52]. When methionine and SAM levels are abundant, SCF^Met30^ binds and ubiquitylates Met4 and Met32. Ubiquitylated Met4 is inactive as a transcription factor [47,53], and ubiquitylated Met32 is marked for degradation by the 26S proteasome [50]. Reduced methionine or SAM lead to dissociation of the ubiquitin ligase SCF^Met30^ from its substrates Met4 and Met32. This in turn blocks substrate ubiquitylation and leads to activation of the transcription factor Met4, and stabilization of the cell cycle inhibitor Mer32. Met4 controls expression of most genes involved in methionine metabolism as well as intersecting pathways, and its activation results in remodeling of metabolic networks to restore SAM levels [54]. Stabilization of Met32 induces a cell cycle arrest in the G1 phase of the cell cycle and a delay in M phase (Figure 4) [44]. The mechanism for M phase delay is not known, but the arrest in G1 appears to be related to destabilized pre-replication complexes, which prevents initiation of DNA replication [44]. Once Met4-directed transcription programs have remodeled metabolic networks to redirect flux to restore SAM levels, the checkpoint arrest is released and cell proliferation can continue. Mechanistically, restored SAM levels promote SCF^Met30^ binding to, and consequently ubiquitylation of, Met4 and Met32, which results in Met4 inactivation and Met32 degradation to terminate the cell cycle arrest. The SCF^Met30^ system provides insight into molecular events of the SAM checkpoint in yeast and shows how methionine metabolism is connected to cell cycle arrest (Figure 5). The human homolog of SCF^Met30^ is the ubiquitin ligase SCF^ßTRCP^. However, SCF^ßTRCP^ has so far not been connected to the methionine dependency of cancer.

Yeast has a second signaling system that responds to methionine and SAM levels. Tu and colleagues found that low methionine levels result in a reduced SAM/SAH ratio [55]. The resulting reduced cellular methylation potential affects carboxyl methylation of the phosphatase PP2A (yeast Pph21 and Pph22). PP2A methylation is necessary to repress autophagy, and low methionine levels are thus inducing autophagy [55]. While the PP2A related pathway is an important sensor for cellular SAM levels, this pathway is unlikely to contribute to the SAM checkpoint in yeast, because the yeast PP2A carboxyl methyltransferase Ppm1 is not required for cell proliferation [56,57]. However, this pathway is important to mobilize nutrients through autophagy to restore methionine levels and cellular methylation potential.

## 7. The SAM Checkpoint in Mammals

Depending on the cancer cell line analyzed, cell cycle arrests in response to methionine restriction or SAM reduction has been reported in the G1, S, or G2 phases of the cell cycle [7,9,13,58,59]. Arrest in G1 has been studied in more detail and is also evolutionarily conserved, because yeast cells induce a robust G1 cell cycle arrest in response to methionine or SAM limitation, whereas G2/M is only delayed under these conditions [44,46] (Figure 4). In both yeast and mammalian cells, pre-replication complexes (preRCs) dissociate from DNA during methionine limitation [7,44]. This effect was monitored by analyzing mini-chromosome maintenance (MCM) proteins, which form the core of preRCs [60]. MCM proteins rapidly redistributed from chromatin associated to soluble fractions without change in their overall abundance, when cancer cells were shifted to homocysteine medium. The MCM loading factor Cdc6, which is necessary for preRC assembly, also dissociated from chromatin. However, in contrast to MCM proteins, Cdc6 overall protein levels were dramatically reduced in homocysteine medium. Surprisingly, neither Cdc6 RNA levels nor protein stability was altered during methionine stress, leading to the proposal that Cdc6 translation is directly affected by shifting cells from methionine containing growth media to homocysteine media [7]. Interestingly, the proposed effect on Cdc6 translation is specific and not part of a general effect on global translation. However, mechanistic insight is currently lacking. Regardless, preRC dissociation can explain the SAM-checkpoint arrest at G1, because preRCs build landing platforms for the DNA replication machinery and are thus essential for the initiation of the *S*-phase (Figure 5).

A second effect on the cell cycle machinery in cancer cells was also observed in response to shifting cells to homocysteine growth media. The activating phosphorylation on threonine 160 (T160) of cyclin dependent kinase 2 (Cdk2) was significantly reduced when methionine-dependent cancer cells were cultured in homocysteine medium. T160 phosphorylation is essential for Cdk2 activity [61]. Accordingly cyclin E/Cdk2 activity was suppressed in homocysteine medium [7]. This suppression of Cdk2 activity further stabilizes the SAM-checkpoint arrest in G1, because cyclin E/Cdk2 is a key component in orchestrating entry into the *S*-phase. Interestingly, Cdk2 association with Cdc6 is important for Cdk2 T160 phosphorylation [62], suggesting that reduced Cdc6 levels may lead to reduced cyclin E/Cdk2 activity, which reinforces the SAM-checkpoint arrest at G1.

Methionine or SAM limitation also results in activation of the mitogen-activated kinase p38 (MAPK14) and its downstream substrate, the MAP kinase MK2 [9]. Chemical inhibition of p38 or MK2 could partially override the SAM-checkpoint arrest, indicating that p38 activation contributes to initiation of the checkpoint program. How p38 activation is connected to Cdk2 phosphorylation on T160 or downregulation of Cdc6 levels is currently not known. However, because p38 or MK2 inhibition can partially overcome the G1 arrest to allow cells to enter the *S*-phase despite reduced SAM levels, the biological importance of the SAM-checkpoint could be evaluated. Consistent with the concept that cell cycle checkpoints are important to protect cells from adverse effects, cells died when the SAM-checkpoint arrest was prevented through p38 or MK2 inhibition [9].

There is strong evidence that methionine dependence of cancer is due to reduced methylation potential when cells are cultured in homocysteine medium. In contrast, normal cells grown in homocysteine medium do not experience such an effect on SAM/SAH ratios, and thus do not induce cell cycle arrest. However, it is important to understand that the SAM-checkpoint program can be triggered in any cell by complete depletion of methionine or inhibition of SAM synthesis [9]. How SAM levels or SAM/SAH ratios are sensed and connected to the SAM-checkpoint is currently not known. Recently, a regulator (SAMTOR) of the mammalian target of rapamycin (mTOR) was identified that binds and inhibits mTOR activity [63]. SAMTOR binding is prevented by SAM, thus connecting SAM abundance to mTOR signaling. However, this pathway is unlikely to be part of the SAM-checkpoint, because several lines of experiments exclude mTOR regulation as part of the cell cycle arrest triggered by the Hoffman effect. For example, mTOR signaling is unaffected when cancer cells are shifted to homocysteine medium, and constitutively active mTOR cannot override the SAM-checkpoint-mediated cell cycle arrest [7,9]. The SAM sensor and signaling components that connect SAM availability to cell proliferation are yet to be identified.

## 8. Methionine Metabolism and Tumor Growth

The Hoffman effect demonstrates that cancer cells and non-tumorigenic cells have different metabolic requirements regarding methionine. Cancer cells rely heavily on exogenous methionine. This increased requirement is also evident in human tumors, which can be readily imaged and differentiated from normal tissue using ^11^C-methionine positron emission tomography (Met-PET) [59]. Especially in glioma, Met-PET imaging performs better than 18F-deoxyglucose PET (FDG-PET), because the elevated overall glucose metabolism in the brain interferes with tumor-specific FDG signals. However, Met-PET has also been evaluated for multiple myeloma and other cancers [64]. Given the evidence that the Hoffman effect is maintained in the context of the whole organism, methionine restriction has been evaluated as a therapeutic approach for cancer. Preclinical models have shown promise with dietary methionine restriction significantly suppressing tumor growth in multiple models, which include both solid tumors and blood cancers [59,65,66,67,68,69]. More extensive studies were done with the fast growing Yoshida sarcomas [70]. A methionine-free diet delayed the onset of tumor growth, but mice eventually showed a significant increase in tumor mass. Surprisingly, once tumors reached a certain size, the onset of tumor regression was observed in mice fed the methionine-free diet. Overall, the response of Yoshida sarcomas to a methionine-restricted diet resulted in increased survival. All mice on regular diets were dead by day 12, whereas all Yoshida tumor-bearing mice survived 30 days, with the last mouse dying at day 38. A methionine-free diet did not have any effect on body weight in these tumor-bearing mice [70]. These are impressive effects induced solely by dietary restriction of methionine. Clinical studies using methionine-restricted diets showed mixed effects, but endpoint data were mainly focused on efficacy of plasma methionine reduction [71]. Plasma methionine levels fell by about 50% and patients lost an average of 0.5kg weight per week. Combination of methionine restriction with 5-fluorouracil in preoperative high-stage gastric cancer patients showed a striking effect on tumor pathology when tumors were examined after surgery [72]. Methionine restriction may thus enhance the response to chemotherapeutics in a synergistic fashion.

To more efficiently reduce plasma methionine than is possible with dietary intervention, a recombinant enzyme that degrades methionine has been developed and produced in *Escherichia coli* [73,74]. The gene was derived from *Pseudomonas putida* and encodes L-methionine-γ-deamino-α-mercaptomethane-lyase, but is usually referred to as methioninase (METase). METase injection showed efficacy in both cell-based and patient derived xenograft (PDX) models of various cancers [69,75,76,77,78,79]. Pilot phase 1 trials administered METase infusions to terminal cancer patients, which had no side effects even though a dramatic reduction of plasma methionine by 200-fold was achieved [59,80,81]. Clinical trials have not advanced past this exploratory state, for reasons unclear to the author.

Methionine restriction in combination with chemotherapy or radiation presents the most promising path to clinical application. Methionine-restricted diets clearly sensitize tumors to chemotherapeutics and radiation [59,82]. Elegant studies linked the increased chemo and radiation therapy response of PDX models fed diets low in methionine to altered flux through the one-carbon cycle, and consequently imbalanced antioxidant and nucleotide metabolism [82]. Importantly, metabolic effects induced by methionine-restricted diets in humans were similar to those that sensitized tumors in mouse models to therapy [82]. These findings suggest that methionine-restricted diets or administration of METase in combination with chemo or radiation therapy could be effective in clinical settings. However, one aspect of methionine restriction should be considered. Methionine restriction leads to reduced SAM levels, and such conditions have been shown to trigger differentiation and apoptosis of pluripotent stem cells [83,84]. Whether similar effects are induced in tissue specific stem cells has not been evaluated, but diet induced differentiation of tissue stem cells could lead to stem cell exhaustion and loss of tissue homeostasis. Exhaustion of tissue-specific stem cells in response to a low methionine diet is, however, inconsistent with the extensive literature showing that methionine restriction expands animal lifespan [85,86,87,88,89,90,91].

## 9. Concluding Remarks

Methionine plays a much more complex role than simply supporting protein synthesis. The metabolic position of methionine links it to epigenetics, nucleotide biosynthesis, membrane lipid homeostasis, and several signaling pathways that are controlled by methylation events. Physiological consequences of methionine imbalance on cancer and aging have been extensively reported, but mechanistic understanding is lacking. It will be essential to identify key players that sense methionine/SAM levels and transmit this information to relevant signaling pathways that impact cell and organismal physiology. Without such understanding, it will be difficult to progress to the clinic. While dietary interventions to reduce exogenous methionine in genetically identical animal models have shown some amazing results in both cancer and aging, the effects of dietary modifications are highly variable in genetically diverse populations such as humans. It is thus important to generate a molecular understanding of the methionine dependence of cancer in order to develop biomarkers to monitor the efficacy of methionine restriction, and to identify potential drug targets that can be exploited to pharmacologically trigger the Hoffman effect.

## Figures and Tables

**Figure 1 biomolecules-10-00568-f001:**
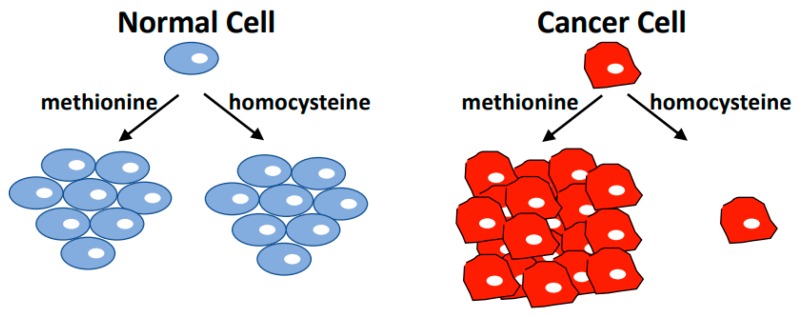
The Hoffman Effect. Non-tumorigenic cell lines have the same proliferation rate in media containing methionine or media where methionine is replaced with the immediate metabolic precursor homocysteine. However, most cancer cells cannot proliferate in homocysteine medium, and induce cell cycle arrest followed by apoptosis when cultured under these conditions. Cancer cells readily synthesize methionine from homocysteine, but appear to depend on exogenously supplied methionine. This cancer-specific metabolic dependence is referred to as the methionine dependence of cancer, the methionine stress sensitivity of cancer, or simply the Hoffman effect.

**Figure 2 biomolecules-10-00568-f002:**
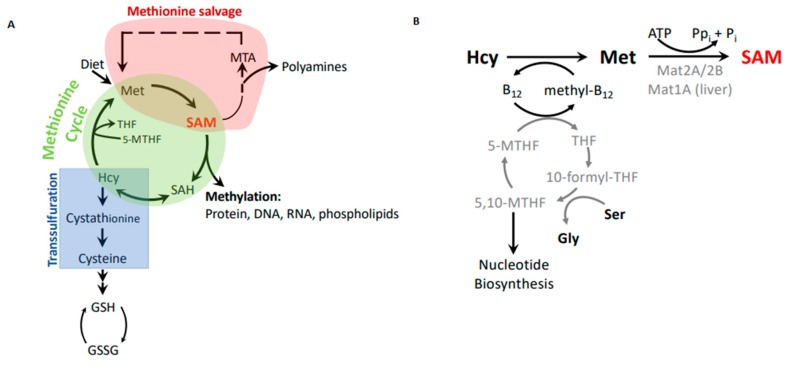
Methionine metabolism. (**A**) Metabolic connections between the methionine cycle, which produces methylation potential, the methionine salvage pathway, which recycles methionine from byproducts of the polyamine synthesis pathway, and the transsulfuration pathway, which generates cysteine and glutathione to combat oxidation. (**B**) The remethylation step of homocysteine as part of the methionine cycle and synthesis of *S*-adenosylmethionine (SAM).

**Figure 3 biomolecules-10-00568-f003:**
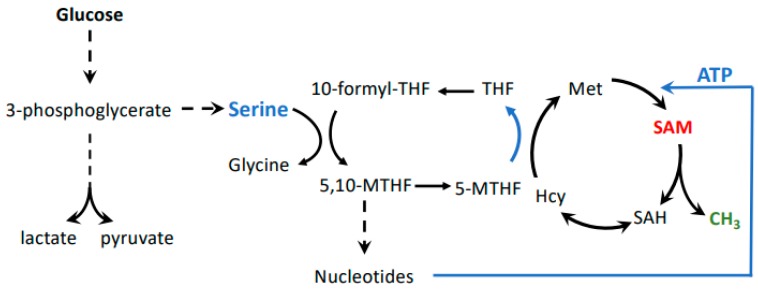
Possible connection between the Warburg and Hoffman effects. Glycolysis is connected to the methionine cycle through the folate cycle, whereby serine derived from 3-phosphoglycerate provides one-carbon units. This pathway is important in cancer cells because the folate cycle feeds both nucleotide and SAM biosynthesis. SAM levels are especially impacted by the folate cycle because it is important for re-methylation of homocysteine to methionine, as well as ATP synthesis. Both methionine and ATP are substrates for formation of SAM.

**Figure 4 biomolecules-10-00568-f004:**
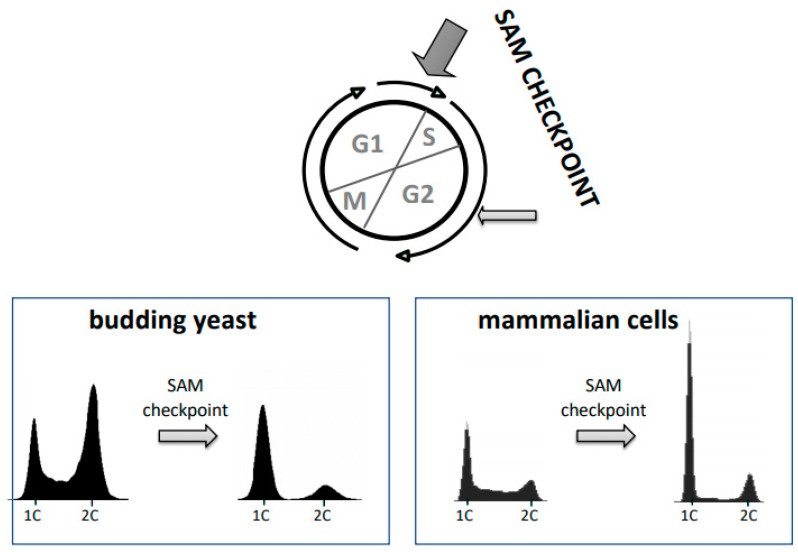
The SAM cell cycle checkpoint. Yeast cells and mammalian cells arrest the cell cycle when SAM levels are limited. This arrest can be induced in yeast and mammalian cells by either SAM limitation or methionine depletion, and in cancer cells also by a shift to homocysteine medium (Hoffman effect). Cells show a robust arrest in the G1 phase of the cell cycle due to lack of stable pre-replication complexes to initiate *S*-phase. A delay in G2/M is also observed. In cancer cell lines S/G2 arrest has also been reported when cells were cultured in homocysteine medium.

**Figure 5 biomolecules-10-00568-f005:**
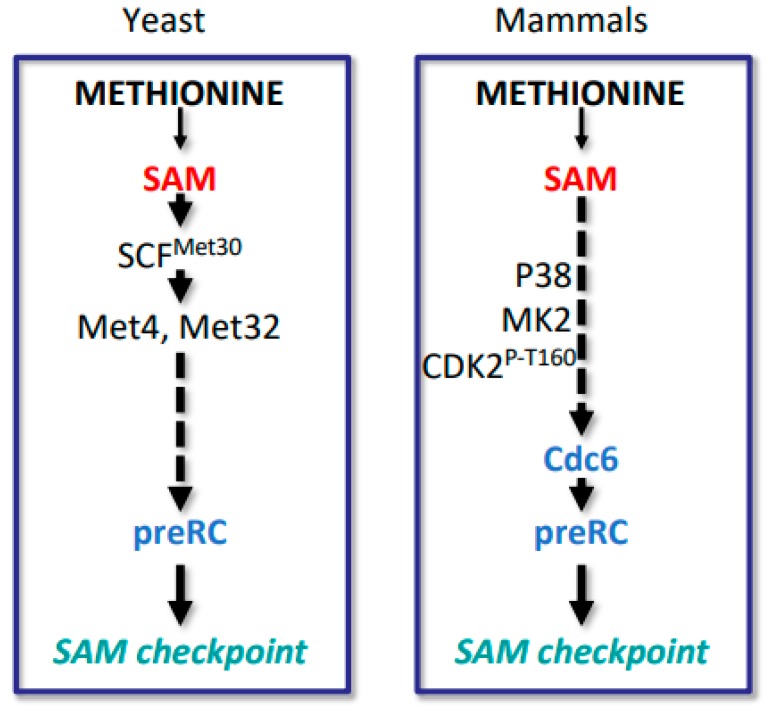
SAM-checkpoint in yeast and mammals. In yeast SAM levels are sensed by the ubiquitin ligase SCF^Met30^, which ubiquitylates several substrates including the transcription factor Met4 to coordinate methionine metabolism with cell cycle control. What senses SAM abundance in mammalian cells in the context of cell proliferation is currently unknown. Both yeast and mammalian cells induce the SAM-checkpoint arrest by destabilizing pre-replication complexes. What components signal SAM levels to pre-replication complex stability is not well understood, but p38, MK2, and activating phosphorylation of T160 of Cdk2 have been implicated.

**Table 1 biomolecules-10-00568-t001:** Cell lines with known growth properties in homocysteine medium.

Cell Line	Methionine Dependence	Tumor Site
MDA-MB468	Yes [7,10]	Breast
MCF7	Yes [5,6,7,10]	Breast
MDA-MB361	Yes [7]	Breast
HCC1806	Yes [10]	Breast
HCC1143	Yes [10]	Breast
SKBR3	Yes [10]	Breast
BT-549	Yes [10]	Breast
ZR-75-1	Yes [10]	Breast
SUM-159	Yes [10]	Breast
T47D	Yes [10]	Breast
W-256	Yes [3,4]	Breast (rat)
MDA-MB231	No [1], moderate [10]	Breast
HCC70	No [10]	Breast
HCC38	No [10]	Breast
SUM-149	No [10]	Breast
MDA-MB231	No [1], moderate [10]	Breast
BxPC3	Yes [11]	Pancreas
PANC1	No [11]	Pancreas
LoVo	Yes [12]	Colon
SK-CO-1	Yes [6]	Colon
PC-3	Yes [5,6,13]	Prostate
LNCaP	Moderate [13]	Prostate
DU145	Moderate [5,6,13]	Prostate
SV80	Yes [3]	Transformed fibroblast
HEK293T	Yes [11]	Transformed kidney cell
W18VA2	Yes [3]	SV40 transformed human cells
J111	Yes [4]	Monocytic leukemia
L1210	Yes [4]	Lymphatic leukemia (mouse)
A2182	Yes [5,6]	Lung
SK-LU-1	Yes [5,6]	Lung
A549	Moderate [6]	Lung
A427	No [5,6]	Lung
J82	Yes [5,6]	Bladder
T24	No [5,6]	Bladder
8387	Yes [5,6]	Fibrosarcoma
HT1080	Yes [5,6]	Fibrosarcoma
HOS	Yes [5,6]	Osteosarcoma
A204	Moderate [6]	Rhabdomyosarcoma
A673	Yes [5,6]	Rhabdomyosarcoma
SK-LMS1	No [5,6]	Leiomyosarcoma
SK-N-SH	Yes [5,6]	Neuroblastoma
SK-N-MC	No [6]	Neuroblastoma
A375	Moderate [6]	Melanoma
MeWo	No [6]	Melanoma
A172	Moderate [5,6]	Glioblastoma
HeLa	Yes [6]	Cervical
A498	Yes [5,6]	Kidney

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
