# Peer review of "Methionine Dependence of Cancer"

_biomolecules, 2020, doi:10.3390/biom10040568_

Round 1

Reviewer 1 Report

This is a nice, concise and interesting review of the field.  It raises many interesting points and appears to be a fair and unbiased review.  I have no issues with this work.  I think it will be of interest to the field.

Author Response

Thank you for reviewing the manuscript. We have made additional changes based on the other reviewers' comments. Changes are indicated in the revised manuscript ("track-changes version)."

Reviewer 2 Report

This is an interesting, informative and very readable review. The illustrations are simple but very helpful.

My only minor suggestion would be that the section on applications in cancer therapy at the end is very brief. I would suggest a little expansion of this section e.g. more details on the preclinical and clinical studies with methioninase would be of interest.

One trivial typo - "provided" rather than "provide" in the abstract.

Author Response

Thank you for reviewing the manuscript. We corrected the typo and expanded on clinical and pre-clinical studies as suggested. Changes are indicated in the manuscript version that includes track changes.

Reviewer 3 Report

In this review the author describes the dependence of cancers on methionine. The author presents a broad overview of the field, explains the underlying mechanisms in some detail and also points out gaps in our understanding and knowledge. Overall, the review is very well written and organized – and was a very interesting read.

Some suggestions/questions that came up while reading the manuscript, which the author might find useful.

  1. In line 123 the author points out that there are several methionine independent cancer cell lines. Given the topic of the review, it would be of interest to the reader if there was some information presented on the cancers that are methionine dependent/ independent. For example, are certain types of cancer prone to being methionine dependent versus not? How would this inform treatment? Maybe this information could be presented as a table.
  1. Line 136: “…loss of tumorigenic properties” It would be helpful to describe what properties are being referred to specifically.
  1. Line 146&147: “… subcellular distribution”; what is the difference in subcellular distribution between exogenous methionine and homocysteine?
  1. Line 166: Is there any information on why homocysteine is shuttled to the transsulfuration pathway in cancers? is the diversion required for tumorigenic properties? what cellular advantage is conferred by shuttling into that pathway?
  1. Line 229: Is there any information on how/why reduced methionine or SAM causes uncoupling of SCF from Met4 and Met 23?
  1. Line 232: “.. cell cycle inhibitor Mer23.” From this sentence it is unclear what exactly Mer23 is an inhibitor of – G1/S phase transition?
  1. In line 234 the author points out that Met32 stabilization is linked to G1 arrest- is there any information on the mechanism linking Met32 to this arrest? Is it through dissociation of the pre-RC? If yes, what is the evidence linking Met32 and pre-RC?
  1. Line 260: “…does not contribute to the SAM checkpoint in yeast…” Is there a reference for this?
  1. Line 275: how were the effects on transcription assessed? Steady state mRNA abundance or were transcription rates directly measured?

Author Response

I would like to thank the reviewer for the important comments. We have made the relevant changes in the manuscript as indicated in the "track-chages" version of the manuscript. We also generated a table with cell lines for which the status of methionine dependence is known.